# Meta Reinforcement Learning for Fast Adaptation of Hierarchical Policies

## Abstract

Hierarchical methods have the potential to allow reinforcement learning to scale to larger environments. Decomposing a task into transferable components, however, remains a challenging problem. In this paper, we propose a meta-learning approach for learning such a decomposition within the options framework. We formulate the objective as a bi-level optimization problem in which sub-policies and their terminations should facilitate fast learning on a family of tasks. Once such a set of options is obtained, it can then be used in new tasks where only the sequencing of options needs to be chosen. Our formalism tends to result in options where fewer decisions are needed to solve such new tasks. Experimentally, we show that our method is able to learn transferable components which accelerate learning and performs better than existing methods developed for this setting in the challenging ant maze locomotion task.

## 1 Introduction

Current state of the art model-free reinforcement learning methods were successfully applied to many challenging tasks [33, 41]. However, one of the main drawbacks of these methods is their data-inefficiency and inability to generalize to other related tasks [12]. It is often impossible to use the agent trained on one task to solve another related task [53] or even to use it as a starting point for training because trained models become increasingly exploitative and thus are unable to explore in a new task. In such cases, we have to gather new data and train a new model which is time-consuming.

One way to mitigate this problem is by learning a policy with reusable modules which can be used in multiple tasks. For example, if we assume that related tasks contain shared sub-tasks (i.e. tasks come from the same family or have hierarchical structure), we can speed up the adaptation to new tasks by learning sub-policies that solve these sub-tasks. This is because solutions to new tasks can be created by combining known solutions to sub-tasks during adaptation. The idea of learning reusable skills in multiple environments, which dates back to at least 1995 [48], was thoroughly explored within the options framework [17, 24, 25, 29, 36, 46].

In this framework, a policy is composed of options (modules that encapsulate sub-policies), and a high-level policy that chooses among them. Options have their own termination function, and a new option is only initiated when the earlier option terminates. Therefore, options define temporally extended behaviors that can form solutions to sub-tasks. Despite extensive research in this area, there is not yet a consensus on answers to many important questions about options: What are good options? How can we find them? When should a termination occur? How many options should one use? In this work our aim will be to find options that allow for fast adaptation to tasks from the same family. We use this single principle to address all of these questions except for the number of options which we consider a hyperparameter.

Learning high-level policy, sub-policies and terminations at the same time is a challenging task. Recent prior work on options proposed a way to learn both, including the terminations, in an end to end manner with policy gradient methods [4, 44]. However, despite achieving good performance in single-task settings, these methods often produce options which may not be useful for transfer [21, 22]. This is because such options are not explicitly trained for multi-task setting and can often terminate too often or not at all [21, 22].

To overcome this issue, Frans et al. [17] proposed to use such method in a multi-task setting with options that have predefined length and are optimized for performance after adaptation of the high-level policy. Although options that terminate after certain amount of steps simplify the problem and work well in some settings [17, 25], manually setting this important hyperparameter requires prior knowledge and might not work in cases where options need to have different lengths [23]. For example, task where this approach would not be preferable could be driving because some driving sub-tasks, such as driving on a highway, are much longer than others, such as driving out from one intersection to another in a city. Consequently, capturing the length of both sub-tasks with a single hyperparameter [17] or range of hyperparameters [25] can become difficult or even impossible. In such cases, learned terminations are preferable.

In this paper, we propose a method for learning options that allows for fast adaptation to multiple tasks. We formalize this notion using recent ideas from gradient based meta-learning [14]. Rather than using options with fixed length [17], our algorithm learns both sub-policies and when to terminate options using a single meta-learning objective. We hypothesize that this objective implicitly encourages options to terminate in a way that yields reusable components. In our experiments, we demonstrate the benefits of our approach in a simple Taxi domain as well as in a complex Mujoco [49] Ant Maze domain [17].

## 2 Related Work

Since our work builds on insights from both hierarchical reinforcement learning and meta-learning, we present related work in both domains separately, in subsections 2.1 and 2.2 respectively.

### 2.1 Hierarchical Reinforcement Learning

One of the aims of hierarchical reinforcement learning is to decompose a complex task or policy into simpler units. Popular approaches include learning a diverse set of skills [11] or utilizing the idea of Feudal Reinforcement Learning [7, 34, 50]. Another large collection of related work instead relies on the options framework [46].

Some works on options rely on so-called bottleneck states that can be used as sub-goals [30, 31, 35] whereas others use spectral clustering to create options [28]. These approaches usually require prior knowledge about the environment which restricts their applicability. Different from aforementioned methods, end-to-end methods such as the ones which rely on the Option-Critic architecture [4, 39] are applicable in more general settings. However, these policy gradient methods can be less efficient than concurrently introduced inference based end-to-end methods [6, 16, 44] because they only update the option that generated the action whereas inference based methods update options according to their responsibilities for each action.

A common problem with end-to-end methods that learn terminations in a single-task setting is option collapse [4]. This causes options to terminate after every action or to never terminate. In such cases the learning of terminations can be facilitated by augmenting the objective with entropy regularization [44] or deliberation cost [21], regularizing towards a termination prior [23], or by optimizing different objective that encourages appropriate terminations [22]. As an alternative, one can also use time-based terminations with fixed [17] or randomized length [25].

### 2.2 Meta-Reinforcement Learning

Meta-reinforcement learning is concerned with producing models which are able to adapt to novel tasks quickly. This sub-field includes a broad range of work such as unsupervised methods [11, 19], methods that rely on latent variables [20, 38] or methods that learn the update rule of a policy [10, 32, 51].

In contrast with the latter, the recent gradient-based method Model-Agnostic Meta-Learning (MAML) [14] assumes that policy parameters are updated with gradient descent and instead aims to learn initial parameter values. MAML was extended in followup works that only trained a part of the network [37, 54] or showed benefits of different architectural choices such as per-parameter learning rates [3, 26]. Several works also focused on MAML in a reinforcement learning setting [2, 27, 45]. In particular, Al-Shedivat et al. [2] and Stadie et al. [45] pointed out a difference between theory and practical implementation of MAML in automatic differentiation frameworks. This issue was further discussed and resolved in followup works [13, 15, 40].

Lastly, there exist methods which do not employ the techniques mentioned above and instead rely on the options framework [5, 17, 23–25, 29, 36, 52] or task-specific policies [47]. These approaches often make different assumptions about the tasks and settings in which they are applied. Some require policies that solve each environment [36] whereas others need environment ID [23, 29] or cumulants that properly represents task dynamics [5]. Closest to our work are Meta Learning Shared Hierarchies (MLSH) [17] and Adaptive Skills Adaptive Partitions (ASAP) [29]. ASAP uses a policy gradient method to optimize immediate performance on multiple tasks with known environment ID but does not use neural networks and does not learn terminations. On the other hand, MLSH uses a hierarchical structure with predefined options length and a problem setting with unknown environment ID. It optimizes for post-adaptation performance by using two alternating phases that either only update high-level policy or both high-level policy and sub-policies simultaneously. This approach does not use the information from the intermediate adaptation steps when calculating the gradient which can negatively affect its accuracy. Additionally, options with fixed length may be difficult to use in some settings as we've described in Section 1.

## 3 Background and Notation

In this section, we will first cover the fundamentals of reinforcement learning, and then focus on the options framework and gradient-based meta-learning.

### 3.1 Reinforcement Learning and the Options Framework

We will consider environments which are episodic Markov decision processes (MDPs). An MDP $\mathcal{M}$ is a tuple $\langle S, A, p_0, P, R, \gamma \rangle$ with $S$ being a set of states, $A$ a set of actions, $p_0(s_0)$ a probability distribution of initial states, $P(s'|s, a)$ a transition probability function, $R(s, a)$ a reward function and $\gamma$ a discount factor.

An agent with a stochastic policy $\pi$ interacts with an environment $\mathcal{M}$ in the following way. At every timestep $t$, the agent receives a state of the environment $s_t \in S$ and selects an action $a_t \in A$ according to conditional distribution $\pi(a_t|s_t)$. Depending on the current state and the action performed, the environment provides the agent with a new state $s_{t+1} \sim P(s_{t+1}|s_t, a_t)$ and a scalar reward $r_t = R(s_t, a_t)$. This process is repeated until a so-called terminal state is reached. We define a trajectory $\tau$ as an ordered sequence of all states actions and rewards in a single episode $\tau = (s_0, a_0, r_0, ..., s_T)$. Similarly, the history at timestep $t$ consists of all states and actions preceding $a_t$, $h_t = (s_0, a_0, ..., s_t)$. The state value function is defined as $V_\pi(s) = \mathbb{E}_\pi[G_t|s_t = s]$ where the discounted return at timestep $t$ is defined as $G_t(\tau) = \sum_{t'=t}^{T} \gamma^{(t-t')} r_{t'}$.

The agent's objective is to maximize the expected discounted return $J = \mathbb{E}_{p(\tau|\theta)}[G_0(\tau)]$. We can maximize the objective with gradient descent by estimating the policy gradient $\nabla_\theta J \approx \mathbb{E}_{p(\tau|\theta)}[\sum_{t=0}^{T} \nabla_\theta \log \pi_\theta(a_t|s_t) A_t]$ using Monte Carlo sampling, where $A_t$ is an advantage estimator such as the generalized advantage estimator $A_t^{GAE}$ [42].

The options framework is a framework for temporal abstraction that consists of options $\omega = \langle \mathcal{I}^\omega, \pi^\omega, \xi^\omega \rangle$ and a policy over options $\pi^\Omega(\omega|s)$. Each option $\omega$ consists of an initiation set, a sub-policy and a termination function. The initiation set $\mathcal{I}^\omega$ is a set of states in which an option can be selected (initiated) and in our case it is the whole state space ($\mathcal{I}^\omega = S$). A sub-policy $\pi^\omega(a|s)$, also called low-level policy, is a regular policy that acts in the environment. Lastly, the termination condition $\xi^\omega(s)$ is a function that outputs the probability of termination for the option in a given state.

## 3.2 Model-Agnostic Meta-Learning and DiCE

Model-Agnostic Meta-Learning (MAML) [14] is a meta-learning technique that trains a model for maximum post-adaptation performance on a distribution of tasks. The adaptation consists of one or several inner gradient updates. If we consider an estimator $f_\theta$ with parameters $\theta$ and a task-specific loss $\mathcal{L}_{\mathcal{M}_i}$, a supervised learning objective with a single inner update can be formalized as shown in Equation 1. In order to optimize this objective one only needs to take a gradient of this expression. This can be easily achieved with automatic differentiation frameworks by creating a backpropagation graph for the gradient.

$$\min_\theta \mathbb{E}_\mathcal{M}\left[\mathcal{L}_{\mathcal{M}_i}(f_{\theta'})\right] = \min_\theta \sum_{\mathcal{M}_i \sim p(\mathcal{M})} \mathcal{L}_{\mathcal{M}_i}(f_{\theta - \alpha \nabla_\theta \mathcal{L}_{\mathcal{M}_i}(f_\theta)}) \tag{1}$$

One can similarly use this approach with a reinforcement learning objective. However, the implementation with an automatic differentiation framework differs because a simple backpropagation through the computation graph of the gradient produces biased gradients [2, 45]. This is due to an additional dependency of the sampling distribution on parameters that is not present in the supervised learning objective. To produce correct higher order gradients with automatic differentiation frameworks in a reinforcement learning setting, one can use the objective in Equation 3 as proposed by Farquhar et al. [13]. This objective utilizes the DiCE operator ⊡ [15] which can be implemented according to Equation 2 where $\perp(x)$ is a stop gradient operator that evaluates to $x$ but returns a zero gradient when differentiated.

$$\boxdot(\boldsymbol{a_t}) = \exp\left[\log \pi_\theta(\boldsymbol{a_t}|\boldsymbol{s_t}) - \perp(\log \pi_\theta(\boldsymbol{a_t}|\boldsymbol{s_t}))\right], \quad \nabla_\theta \mathbb{E}_{\tau \sim p(\tau|\theta)}\left[G_0^{\mathcal{M}_i}(\tau)\right] \approx \nabla_\theta J_\boxdot \tag{2}$$

$$\nabla_\theta J_\boxdot = \mathbb{E}_{\tau \sim p(\tau|\theta)}\left[\sum_{t=0}^T \nabla_\theta \left(\prod_{t'=0}^t \boxdot(\boldsymbol{a_{t'}})\lambda^{t-t'} A_t^{GAE} - \prod_{t'=0}^{t-1} \boxdot(\boldsymbol{a_{t'}})\lambda^{t-t'} A_t^{GAE}\right)\right]. \tag{3}$$

## 4 Fast Adaptation of Modular Policies

Much of the extensive research in the options framework has focused on an intuition of options capturing useful sub-tasks [4, 17, 36, 46]. However, there is no consensus about capturing this intuition in an objective function or the best way to find such options. We propose a conceptually simple objective: a good set of options allows quick adaptation to many novel tasks. This can be formulated using the MAML framework [14], where we consider a setting in which there is a distribution of tasks $p(\mathcal{M})$ with similar (hierarchical) structure but different reward or transition functions. Our goal is then to maximize the expected performance after $L$ adaptation steps of the hierarchical policy parametrized by $\theta$:

$$\max_\theta \sum_{\mathcal{M}_i \sim p(\mathcal{M})} \mathbb{E}_{\tau^L \sim p(\tau^L|\theta^L)}\left[G_0^{\mathcal{M}_i}(\tau^L)\right], \quad \theta^{j+1} = \theta^j + \alpha_{in}\nabla_{\theta^j}\mathbb{E}_{\tau^j \sim p(\tau^j|\theta^j)}\left[G_0^{\mathcal{M}_i}(\tau^j)\right]. \tag{4}$$

Using conventional MAML means adapting a large number of parameters which can be disadvantageous, as was demonstrated by Zintgraf et al. [54] and Antoniou et al. [3]. By reducing the number of parameters that are tuned during the adaptation phase, one can reduce the complexity of the problem during test time at the cost of a less expressive policy. We thus split the parameters into an inner group $\theta_{\text{in}}$ and an outer group $\theta_{\text{out}}$ where inner parameters are updated during the adaptation step and outer parameters are optimized in the outer objective. Note that when using such split, the initialization values of inner parameters may also be meta-learned [54]. We experimented with both versions and observed that fixed initialization values performed better. Similarly, the per-parameter inner learning rate $\alpha_{in}$ [3, 26] can also be meta-learned to allow for more complex inner updates. We used this approach in a setting with more complex environment.

Our option model has three sets of parameters: those of the high-level policy network $\theta_\Omega$, sub-policy networks $\theta_\omega$ and termination networks $\theta_\xi$. We now divide these over the inner and outer parameter group. Since we assume that tasks with common sub-problems can be solved using identical options, we consider the sub-policy and termination function parameters as outer parameters. On the other hand, since in each task the decision of the high-level policy to choose options would be different, its parameters constitute the inner group. By keeping sub-policies fixed during the adaptation and

---

**Algorithm 1** Fast Adaptation of Modular Policies

---

initialize $\theta_\Omega, \theta_\xi, \theta_\omega, \alpha_{in}, \alpha_{out}$
set $\theta_{in} = \theta_\Omega$
set $\theta_{out} = \{\theta_\xi, \theta_\omega\}$
**repeat**
   Set gradient of outer parameters $\boldsymbol{g}_{\theta_{out}} = 0$
   **for** $n = 1$ **to** $N$ **do**
      set $\theta'_{in} = \theta_{in}$
      sample a task $\mathcal{M} \sim p(\mathcal{M})$
      **for** $l = 1$ **to** $L + 1$ **do**
         sample $k$ episodes $\tau_{1:k}$ on $\mathcal{M}$ using $\pi_{\{\theta'_{in},\theta_{out}\}}$
         fit a baseline $V_\kappa$ using data from $\tau_{1:k}$
         compute $A_t^{GAE}$ for all $\tau_{1:k}$
         compute $\log \pi(\boldsymbol{a_t}|\boldsymbol{h_t}) = \mathbb{E}_{\omega|\boldsymbol{h_t}}[\pi^\omega(\boldsymbol{a_t}|\boldsymbol{s_t})]$
         compute $J_{\square}$ with $A_t^{GAE}$, $\log \pi(\boldsymbol{a_t}|\boldsymbol{h_t})$ (Eqs. 2, 3)
         **if** $l < L + 1$ **then**
            $\theta'_{in} = \theta'_{in} + \alpha_{in}\nabla_{\theta'_{in}}J_{\square}$
         **else**
            $\boldsymbol{g}_{\theta_{out}} = \boldsymbol{g}_{\theta_{out}} + \nabla_{\theta_{out}}J_{\square}$
   $\theta_{out} = \theta_{out} + \alpha_{out}\frac{1}{N}\boldsymbol{g}_{\theta_{out}}$
**until** convergence

---

restricting the update to the high-level policy, we optimize for options that can be used to solve multiple tasks, thereby allowing the overall policy to adapt quickly with the change of high-level policy. This also allows for an expressive policy which can capture different behaviors and reduces the number of parameters and decisions an agent needs to learn and make during test time.

Formally, our final objective can be expressed as Equation 5 with the inner update given by Equation 6. The objective is similar to the one used in MLSH [17] with some key differences. Firstly, by backpropagating through the update step we are able to capture additional information from the adaptation steps in the gradient and secondly, our objective includes the optimization of termination parameters and thus allows for options with different lengths.

$$\max_{\theta_\omega, \theta_\xi} \sum_{\mathcal{M}_i \sim p(\mathcal{M})} \mathbb{E}_{\tau \sim p(\tau|\{\theta_\omega, \theta_\xi, \theta_\Omega^L\})} \left[ G_0^{\mathcal{M}_i}(\tau) \right] \tag{5}$$

$$\theta_\Omega^{j+1} = \theta_\Omega^j + \alpha_{in}\nabla_{\theta_\Omega^j}\mathbb{E}_{\tau \sim p(\tau|\{\theta_\omega, \theta_\xi, \theta_\Omega^j\})} \left[ G_0^{\mathcal{M}_i}(\tau) \right]. \tag{6}$$

## 4.1 Algorithm

Written in its general form the objective leaves some freedom with regard to which policy gradient algorithm is used for gradient calculation. In our work we use the Inferred Option Policy Gradient (IOPG) [44] because it updates all options at the same time based on their responsibilities, i.e., the probability that the option was active given the history $\boldsymbol{h_t}$ of states and actions so far. This can lead to better data-efficiency when compared to other methods that only update a single option at a time but comes at the cost of increased computation time. Another important design choice is the state value function estimator. In the MAML RL setting the policy constantly changes in every inner update. It is thus difficult to use past trajectories for fitting the value function. We therefore use a linear time-state dependent baseline [9] which works better than more complex baselines with little data and was also used in the original MAML implementation.

The resulting algorithm for Fast Adaptation of Modular Policies (FAMP) is outlined in Algorithm 1. Note that in order to use IOPG with DiCE we replace $\pi(\boldsymbol{a_t}|\boldsymbol{s_t})$ with $\pi(\boldsymbol{a_t}|\boldsymbol{h_t})$ in Equation 2. An intuition about why this is possible comes from the fact that we can easily formulate a new MDP $\tilde{\mathcal{M}}$ in which states $\tilde{s}_t$ are histories $\boldsymbol{h_t}$ of the original MDP without otherwise altering the dynamics. After $L$ inner updates, the gradient of the objective with respect to the outer parameters is calculated. In principle, we would like to optimize for performance after a moderate number of gradient updates $L$ such as 10 or 20. However, with more inner updates the resulting gradient of the objective becomes

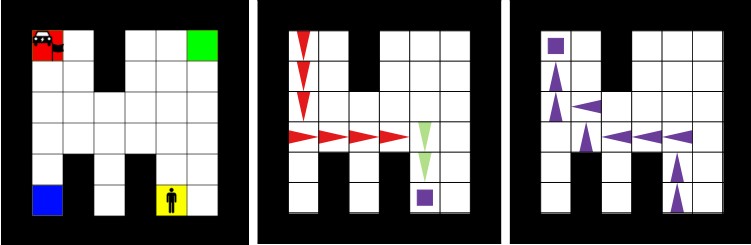

Figure 1: *Left:* Map of a taxi environment with special states and an example task. *Middle and Right:* Visualization of the option usage in this task. Middle part shows states without passenger on board. Right part shows states with passenger. Arrows represent directional actions, pick-up/drop-off is shown as a square. Each action is colored according to the active option.

noisier due to the usage of Monte Carlo estimate in each inner update. Furthermore, the time complexity of gradient computation and sample complexity both scale linearly with the number of inner updates. In practice we found a range from 2 to 4 update steps to be acceptable. An important benefit of gradient-based meta-learning is that even though the model is optimized for performance after $L$ adaptation steps, it can still be improved after $L$ updates by performing more steps of gradient descent.

## 5 Experiments

In this section, we empirically evaluate our method and show its benefits when applied to randomly selected tasks within and outside of the training distribution.

### 5.1 Taxi

In the first set of experiments we use a modified Taxi environment[1] [8] displayed in Figure 1. An agent acts as a taxi driver who starts in one of the special (colored) locations. The goal of the driver is then to drive a passenger from one of the special locations to his destination. The task family consists of 60 different tasks with different combination of start, goal and passenger positions. These are always initialized in special states. In 12 out of 60 easier configurations the passenger starts the episode in the car. The only restriction on start, goal and passenger positions in all cases is that passenger's destination must not be the same as his initial position. Each task is an MDP in which the agent can use 4 directional actions and two special actions: pick-up/drop-off and no-op. The state space is represented as a one-hot vector with 72 entries for every combination of possible taxi location and passenger being on board. Thus *the agent does not have any information about the location of the passenger or goal state*. Therefore, in order to facilitate fast adaptation to the (unobservable) passenger and goal locations, the agent must acquire options that can serve as building blocks for exploration. The reward is 2 for reaching the goal and $-0.1$ per step otherwise. To speed up training in the early phases, we terminate the episode if it takes longer than 1500 timesteps.

In this experiment, we use tabular representations implemented as a combination of linear layer and non-linearity for the policy over options, terminations and sub-policies such that each one-hot state has its own set of parameters. We use 48 training tasks to train sub-policies and terminations with our algorithm. Learned terminations and sub-policies are then kept fixed during test time and only the policy over options is updated. Performance is then compared on the remaining 12 test tasks (selected to use combinations of special locations with similar frequency) to MLSH and two baselines. We chose MLSH because it is a closest hierarchical method designed for our setting in which there is no extra information about the environment available. This is in contrast with many other hierarchical [5, 23, 29] and non-hierarchical [38, 47] meta-reinforcement learning methods which utilize extra information such as the ID of a sampled environment.

Similarly to our method, *MLSH* is trained on all training tasks and evaluated with fixed sub-policies. The *multi-task* baseline is an IOPG algorithm that learns a shared policy (including high-level policy)

---
[1]Details are included in Appendix B

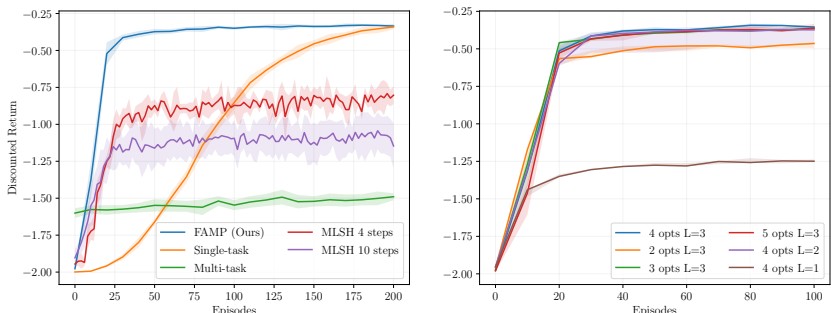

Figure 2: *Left:* Average performance of different algorithms on taxi environment test tasks. Plot shows mean and standard deviation over 5 seeds. *Right:* Average performance of our method with different hyperparameter values on taxi environments test tasks. Plot shows median and interquartile range over 5 seeds.

by optimizing average return over tasks rather than the meta-learning objective in Equations 5 and 6. After the training, it only adapt its high-level policy on test tasks. We expect this baseline to perform poorly in the long run because it does not optimize for post-update performance. Lastly, the *single-task* baseline is an IOPG algorithm that learns the test tasks from scratch without any pre-training. Therefore, since it does not need to generalize to many tasks and has a policy with enough capacity, we expect that it should eventually outperform other methods after sufficiently long training. However, meta-learned policy with desirable options should find good solution much quicker. To make the single-task baseline as strong as possible, we set its learning rate to the highest value that was able to solve all tasks reliably.

**Results**

As shown in Figure 2, our method is able to outperform both MLSH and the multi-task baseline reaching the final performance of $-0.315$. Furthermore, it also outperforms all other algorithms in terms of adaptation speed. We additionally checked whether the single-task baseline eventually overtakes FAMP and found that after more than 200 episodes, its performance stabilizes at a final discounted return value of $-0.284$. This demonstrates that FAMP can learn sub-policies and terminations that allow for fast adaptation in similar unseen environments at the cost of slightly lower asymptotic performance. An example trajectory that was produced by the agent in one of the hardest test tasks is displayed in Figure 1. In this task, the agent is able to combine three options to form an optimal solution. Plots with meta-training curves and learned options are included in Appendix C.

In Figure 2 (right), we show how the performance varies with changes to important hyperparameters, namely, the number of options and adaptation steps. We observe that decreasing the number of adaptation steps during training to one leads to a clear drop in performance. This can be attributed to the policy not being able to switch from exploratory to exploitatory behavior in a single inner update as well as the smaller amount of data observed before each outer update.

Unlike the number of adaptation steps, the number of options does not seem to affect the performance too much. The only noticeable exception is lower performance when using only 2 options. This exception can be explained by noticing that in some states one needs to perform 3 different actions to represent all optimal paths. As an example, consider the state two squares above the blue special state in Figure 1. To reach the blue state in the minimum number of steps the agent needs to use the down action. Similarly, to go from the blue state to the red or yellow one it needs to use up and right respectively. Thus the agent cannot represent the optimal policies with only 2 options. Interestingly, even in this case, the agent is still able to separate trajectories in such a way that it can reach all goals albeit with slightly worse performance.

This outcome demonstrates another benefit of learned option lengths as the optimal option length does not only depend on tasks and their difficulty but also on the number of options that are available. To illustrate this, consider an extreme case where there are as many options as tasks. In this case, it would be sensible to have solution to a different task in each option and not terminate at all because each task would be solved with only one high-level action. However, as the number of available options decreases, sharing options between tasks becomes necessary and terminations should start to

Table 1: Percentage of terminations in trajectories obtained from adapted policies averaged over 5 seeds. Standard deviations are in 1-2% range.

| Number of options | 2 | 3 | 4 | 5 | 8 | 16 |
|---|---|---|---|---|---|---|
| Avg. terminations in traj | 70% | 63% | 57% | 55% | 44% | 28% |

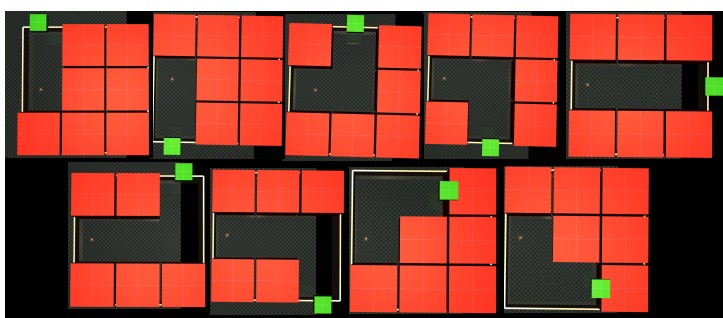

Figure 3: Ant maze tasks. The agent needs to control a simulated 4-legged ant-like robot and move it towards the green square.

occur to allow for all tasks to be solved. Moreover, if the number of options is decreased even further, there may not be enough to options to capture the optimal behavior for all tasks. Consequently, it becomes even more difficult to choose the appropriate option length a priori since it can depend on the number of available options. To confirm this intuition, we ran a followup experiment with longer time horizon in the taxi environment. While the trajectories produced by adapted policies had similar length, relative number of terminations decreased with the increase in the number of options as shown in Table 1.

## 5.2 Ant Maze

In our second experiment, we demonstrate the applicability of our method to more complex environments. We use the family of ant maze tasks introduced by Frans et al. [17] shown in Figure 3. This allows us to reproduce the results of MLSH as closely as possible by mirroring the setting used in the original paper. In addition to MLSH, we also compare to $RL^2$, a non-hierarchical meta-learning algorithm designed for fast-adaptation and Proximal Policy Optimization (PPO) [43], which serves as a strong single-task baseline.

In each task the agent needs to move a simulated 4-legged ant-like robot through a small maze towards the goal. Both state space and action space are continuous with 29 and 8 dimensions respectively and each episode lasts 1000 timesteps. *States do not contain any information about the maze layout or the location of the goal*. The original implementation also resets the orientation of the ant every 200 steps. However, we removed these resets because they made the MDP partially observable, introduced discontinuities and were not realistic for the robotics scenario they are supposed to imitate. Results of experiments with the original implementation are similar to the ones we present. They can be found in Appendix C along with meta-training plots.

Both FAMP and MLSH use the same architecture with two hidden layers of 64 nodes to represent the high-level policy, sub-policies and terminations (only applies to FAMP). We used existing repositories for the implementation of $RL^2$ [18] and PPO [1]. Hyperparameter values can be found in Appendix B. During the training phase, sub-policies (and terminations) of both hierarchical algorithms were trained on all tasks until the return averaged over all environments stopped improving. In the test phase all parameters except for the policy over options were frozen. Similarly, $RL^2$ was pre-trained on all tasks and subsequently evaluated while PPO was trained from scratch.

The comparison of the performance and speed of adaptation can be seen in Figure 4 (left). Our method achieves superior performance reaching an average return of 1330. We also observed a similar trend across individual environments. Plots of these comparisons are available in Appendix C. While the zero-shot performance of $RL^2$ is slightly better than FAMP, it often struggles to further adapt to specific tasks and quickly gets outperformed by both hierarchical methods. This is likely be

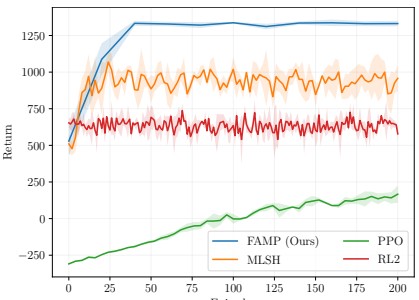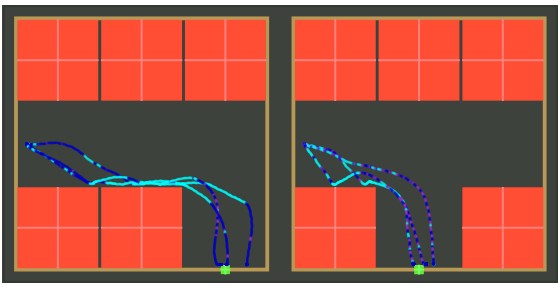

Figure 4: *Left:* Average performance of algorithms on ant maze environments tasks. Plot shows mean and standard deviation over 3 seeds. *Right:* Option usage visualization on ant maze tasks. Both plots were created using positions of the ant during 3 trajectories. Each of the 3 options is represented by a different color.

due to the objective that optimizes average return over all training episodes and not post-adaptation performance directly. Lastly, PPO continuously improves but its performance does not come close to the meta-learning algorithms. After about 1000 episodes it reaches the performance of MLSH and if ran sufficiently long , we would expect that it would eventually catch up to FAMP.

We visualize the option usage of FAMP on two example tasks in Figure 4 (right). After the high-level policy is fine-tuned, we use the $x$ and $y$ positions of the ant in 3 sampled trajectories to highlight which option is active at each part of the state space. Although we only take 2 out of 29 dimensions into account, we are still able to get useful insight about the learned option structure. In the task that is depicted in the left part of the plot, the agent uses the blue option before switching to cyan in the middle and finishing with a combination of blue and purple. On the other hand, in the right task, the agent uses a combination of blue and purple to move down instead of to the right. This shows that the agent learned a useful abstraction that allows it to perform two different useful behaviors in similar parts of the state space by using terminations and different options.

# 6 Discussion and Future Work

In this work, our aim was to learn both sub-policies and terminations of options by using a single simple principle: options should accelerate adaptation in many tasks. We proposed a method for learning hierarchical policies that combines the options framework with gradient-based meta-learning and explicitly optimizes for performance after several adaptation steps. In our experiments, we have demonstrated the benefits of our approach in quickly learning previously unseen test tasks. Furthermore, we have shown that the proposed method outperforms the closest hierarchical and non-hierarchical meta-reinforcement learning methods designed for similar setting in a challenging multi-task learning scenario.

The computation limitations of our method are mostly connected to the calculation of responsibilities in IOPG. In this calculation, many sequential matrix multiplications are required both in the forward and backward pass. The compute time for each update is thus dependent on the trajectory length because these calculations cannot be done in parallel. One direction for future work could thus be alleviating this limitation.

Our objective does not explicitly constrain the number of terminations as long as they lead to fast adaptation. Thus, there are many combinations of options with different lengths which can lead to good performance on all tasks, which do not always correspond to intuitive decompositions. One possible cause of spurious terminations lies in the continuous state space used in some tasks. When neural networks are used to represent termination functions, they learn to generalize to nearby states. In tasks such as the ant maze, the agent will visit many states in the same neighborhood and might thus terminate options several times in quick succession. A promising topic for future investigation is whether this problem could be alleviated by using terminations that also depend on the state in which the option was initiated.

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
