# OpenReview forum: "Meta Reinforcement Learning for Fast Adaptation of Hierarchical Policies"
_NeurIPS.cc/2021/Conference — NeurIPS 2021 Submitted_

### Official Review · Reviewer_JWbv · 2021-07-14

**Rating:** 6
**Confidence:** 4

**Summary:**

This paper proposes to use meta reinforcement learning in order to find a higher-level policy that can quickly update to achieve higher rewards and new tasks. The method uses the options framework in this case and uses meta-learning to find and quickly update the option selection policy given only a little bit of data from the environment. The method does show some improved performance using this particular structure over two different environments, one that was used in the previous work.

Pros
- The Meta-learning structure that's described in the paper does appear to be rather effective and somewhat novel. It shows some similarities to having a high-level policy that is more of a latent policy in this case in the middle learning performs a gradient step in order to adjust which options end up getting selected for a particular task. But the overall method is different enough from prior methods.
- The described method in the paper is also not significantly complicated and seems to fit in fairly well to previous hierarchical reinforcement learning designs in particular for using an options type setup that includes a termination function.


Cons
- While the results look promising they are applied to a limited number of environments. One environment being a discreet environment and the other being the environment from the MLSH paper. Well, potentially not necessary it could still help bolster the thoroughness of the findings in the paper if the evaluation is performed over additional environments. For example using some of the environments from meta world or other hierarchical reinforcement learning papers.
- Well the paper does outline its novel design structure for applying the meta-adaptation step on the higher-level policy there isn't any ablation analysis in the paper to really understand the most important components of the overall method. For example, additional analysis that includes all of the levels of the hierarchy inside of the inner meta adaptation step would improve the clarity of the experiments and the solidity of the findings. Overall the paper seems to be on solid ground but the additional analysis would add to the contributions of the paper.

**Main Review:**

Additional comments:

- It is interesting that the analysis in the paper indicates that the method's performance does not significantly depend on the number of options that are available to the agent. This is usually an issue for these types of methods. It would be helpful to provide some discussion on why the method does not suffer from this common issue.
- RL squared algorithm does not show any improvement in the evaluation environments. That meta-learning algorithm has been evaluated on similar tasks to this ant maze environment and has shown some success. It is possible this baseline is not applied correctly in the work.

----- Post Author Feedback -----
Given the author's feedback, my rating for the paper will stay the same.

**Time Spent Reviewing:**

2.5

---

> ### Author Response · Authors · 2021-08-10
> **Response to Reviewer JWbv**
>
> Thank you for your constructive comments. As was pointed out, the performance of our method indeed seems to be less dependent on the number of used options when compared to methods that only update an active option such as MLSH. This is likely because we use IOPG which updates *all* options according to their responsibilities and not only the option that performed the action. The learning of options is thus more data-efficient when compared to methods that only update the option which was active. Furthermore, as shown in Fig. 5a of [6], IOPG itself is also not very dependent on the number of options as long as there is a minimum number of options to do the task.
>
> Initially, we were also surprised by the performance of RL$^2$ baseline. However, after searching for plausible hyper-parameters in other works, we couldn't find a work where this algorithm works well in Mujoco locomotion with longer horizon. In particular, in [1,2], the algorithm does performs poorly in similar envs and in [3], the results are very similar to ours where it performs reasonably well but struggles with test time adaptation. It is unlikely but not impossible that we missed some work where RL$^2$ worked well in such setting. If you are aware of such work, could you please provide a reference?
>
> We see how the inclusion of a small ablation study where different/all components are adapted in the inner update can improve the clarity of the experiments. Since this does not require any significant changes in the algorithm design or code and this ablation was unanimously proposed by all reviewers, we will include this ablation study in the final version of the paper. However, we do not expect this version to work as well because such version was mostly successfully applied on locomotion problems with shorter time horizon and lower complexity [4,5]. In our case, the horizon is much longer (1000 timesteps vs 100-200) and the environment requires more complex behaviors to be solved due to walls.
>
> [1] Meta-Reinforcement Learning of Structured Exploration Strategies (Gupta et al. 2018)
> [2] Efficient Off-Policy Meta-Reinforcement Learning via Probabilistic Context Variables (Rakelly et al. 2019)
> [3] A Simple Neural Attentive Meta-Learner (Mishra et al. 2017)
> [4] ProMP: Proximal Meta-Policy Search (Rothfuss et al. 2019)
> [5] Model-Agnostic Meta-Learning for Fast Adaptation of Deep Networks (Finn et al. 2017)
> [6] An Inference-Based Policy Gradient Method for Learning Options (Smith et al. 2018)

---

### Official Review · Reviewer_NbWu · 2021-07-16

**Rating:** 4
**Confidence:** 4

**Summary:**

This paper proposes an option method where the high-level agent is a meta-learning agent that can fast learn to select proper options through sampled few episodes. DiCE is used to correct the gradients. The paper demonstrates the effectiveness of the approach on Taxi and Ant Maze environments and outperforms RL^2 and PPO baselines.

**Limitations And Societal Impact:**

Yes, the author discussed the limitations of computation.

**Main Review:**

The paper's presentation is very clear and I appreciate the idea to combine meta-learning and the options framework, but I do not think that the experiments fully demonstrate the potential of the method. First of all, the experiments in the paper focus on navigation. Ant Maze environment is not a challenging task for other hierarchical RL methods. The difficulty comes from forbidding the agent from observing the maze layout. However, since the maze size is rather small, the number of possible layouts is also small. It is possible that that the agent simply remembers the solution to all map layouts and uses the meta-learning trajectories to identify the maze layout. The experiments do not demonstrate the generalizability of the novel map layout. In this case, why not simply learn a recognition model to reconstruct the maze layout [1] and then learn a goal-conditioned policy? The approach is not significant unless it can generalize to a novel layout in a larger maze. Otherwise, I would doubt that we should use search instead of meta-learning for solving complex planning tasks. Besides, since the number of options is small, and the results are not very interpretable, if we need options is questionable to me. The paper lacks a MAML-like meta-learning baseline to demonstrate that options are necessary.

Besides, I am confused by the plots in the paper. Fig 4 shows that FAMP reaches 1200+ score after 40 episodes, but according to the pseudocode and Table 2, in each update step of the meta-agent, it needs to sample N\times L+1\times k episodes, which is larger than 40. Does this mean that there is no need to learn the meta-agent?

In short, the paper has good motivation, however, the experiments have flaws and are not strong enough to demonstrate the potential of their approach. I hope the authors can apply their approach to more complex domains and show its significance.

[1]Multi-task Batch Reinforcement Learning with Metric Learning

**Time Spent Reviewing:**

4

---

> ### Author Response · Authors · 2021-08-10
> **Response to Reviewer NbWu**
>
> Thank you for your review and feedback. Figure 4 shows the adaptation during test time after the model was trained with Algorithm 1. During training, the algorithm indeed uses $N\times(L+1)\times k$ episodes per epoch. However, during test time it only adapts the high-level policy with (inferred options) policy gradient using $k$ (in this case 20) episodes per epoch. We would like to emphasize that we focused on navigation tasks because they naturally combine with our assumption that tasks contain shared sub-tasks (have hierarchical structure). While the application to other environments such as MetaWorld would be interesting, these would be less compatible with our method because there is less overlap between sub-tasks.
>
> Regarding generalization to the novel map layouts, while indeed we didn't separately evaluate this for the ant maze environment in the paper, we did test this for the taxi environment as described on page 6 with results shown on page 7. These results show good performance for generalization to novel maps in the taxi domain at meta-test time.
>
> To further support this claim, we measured the adaptation performance on 4 new tasks in the AntMaze domain that was used in the paper for a clear comparison to MLSH. Although it was not possible to create a new 3x3 test maze that only contains sub-paths from training environments (as was done in the experiment with Taxi), we used 4 layouts (shown below) that share part of the optimal trajectory with original 9 tasks. We obtained a post-adaptation performance (table below) that is similar to the one in Fig 4 when using pre-trained models (trained on original 9 tasks). This shows that the models trained by our method can also generalize to new tasks with similar hierarchical structure in the AntMaze domain.
>
> It is indeed the case that not knowing the layout or a task index is what makes the ant maze task harder than superficially similar tasks in the HRL literature. While theoretically our architecture could represent a pathological solution that represents each particular AntMaze using a pre-learned option policy ('remembering' the solutions), in the qualitative analysis of the learned strategies we see that this is not what is typically learned in practice. Furthermore, even in the case in which the algorithm would learn such pathological solution, it would still need to learn a complex underlying structure in order to adapt to all tasks quickly. While it is entirely possible that learned recognition model and goal-conditioned policy or other multi-task algorithm could perform similarly or better on AntMaze (multi-task learning), these methods would not be applicable in the meta-learning setting (Taxi) which is the main focus of our work.
>
> Lastly, we see how the inclusion of MAML baseline can improve the clarity of the experiments. Since this does not require any significant changes in the algorithm design or code and this ablation was unanimously proposed by all reviewers, we can include this ablation study in the final version of the paper. However, we do not expect this version to work as well because such version was mostly successfully applied on locomotion problems with shorter time horizon and lower complexity [4,5]. In our case, the horizon is much longer (1000 timesteps vs 100-200) and the environment requires more complex behaviors to be solved due to walls.
>
> Additional AntMaze environment layouts (X - block, S - start, G - goal). Structure is similar to Fig 3. with blocks at different places:
> ```
> +---+---+   +---+---+   +---+---+   +---+---+
> |     G |   | X X X |   | X   G |   | X X X |
> | S X X |   | S X X |   | S   X |   | S   X |
> | X X X |   |     G |   | X X X |   | X   G |
> +---+---+   +---+---+   +---+---+   +---+---+
> ```
> Adaptation results on 4 new maze layouts. Setting is identical to that in Fig 4 (same pre-trained models) but with unseen tasks:
> ```
> +---------+-----+-----+-----+-----+-----+-----+-----+-----+-----+-----+-----+
> |Episodes |    0|   20|   40|   60|   80|  100|  120|  140|  160|  180|  200|
> +---------+-----+-----+-----+-----+-----+-----+-----+-----+-----+-----+-----+
> |Return   |  -60| 1732| 1837| 1812| 1820| 1804| 1798| 1834| 1875| 1861| 1816|
> +---------+-----+-----+-----+-----+-----+-----+-----+-----+-----+-----+-----+
> ```
>
>
> [4] ProMP: Proximal Meta-Policy Search (Rothfuss et al. 2019)
> [5] Model-Agnostic Meta-Learning for Fast Adaptation of Deep Networks (Finn et al. 2017)

---

### Official Review · Reviewer_uSF5 · 2021-07-26

**Rating:** 6
**Confidence:** 4

**Summary:**

The paper presents a meta-learning approach for learning hierarchically structured policies, where the agent is trained on some distribution of similar tasks to maximize the post-adaptation performance, as well as learning useful sub-policies in different time scales that enable fast adaptation. The overall model contains a high-level policy selecting over a set of options, and each option contains a sub-policy controlling action selection and a termination probability controlling the duration of low-level policy execution.  During training, the authors argue that the update for low-level policies and termination probabilities should be slower than the update for the high-level policy, to potentially encourage learning sub-policies that can be used in multiple tasks. This is the key idea of the proposed fast adaptation of modular policies (FAMP) algorithm. The experiments on Taxi and Ant Maze environments show FAMP can fast adapt and achieve better returns than other baseline meta-learning methods on unseen test tasks.


**Limitations And Societal Impact:**

The authors discuss the limitation of the non-parallelizable responsibilities calculation of IOPG. I also don't foresee any negative social impact of this work.

**Main Review:**

The paper tries to address an important problem in RL to learn hierarchical policies that enable better adaptation. The proposed algorithm follows the options framework and utilizes recent optimization methods, including MAML and DiCE. The paper has good coverage of related work and is distinguished from existing work by learning both sub-policies and termination conditions. Regarding experiments, the adaptation results look promising, and the selection of baselines is reasonable because MLSH does not utilize the environment id.

However, a few designing factors remain unjustified in the current paper. More ablation study is required to understand their effects, in addition to the comparison with MLSH. First, one key contribution, as claimed, is more flexible termination of sub-policies. However, it is unclear about its actual performance gain and how fixed termination would affect the learning sub-policies, and the adaptation performance remains to be checked. Second, the paper argues that fixing sub-policy in the inner-loop encourages better sub-policies. However, this intuition is not yet supported by any quantitative results.

Also, it seems that FAMP does not require warm-up for sub-policies, while MLSH needs warm-up. Perhaps a related discussion on this is needed. How are sub-policies and termination conditions initialized? If the initial performance of sub-policies is undesired, fixing sub-policies in the inner-loop may lead to slower meta-learning.

Overall, the paper presents an interesting idea for hierarchical RL, and experiments are sufficient to prove its effectiveness in terms of better adaptation performance. However, the learned sub-policies on the Taxi task seem not easy to be interpreted. Moreover, IOPG may be computationally heavy. More analysis would strengthen the paper in understanding the effects of key designing components.

Typos:
line 124, definition of $G_t$ should be $G_t(\tau) = \sum_{t’=t}^{T}\gamma^{(t’-t)}r_{t’}$.
Eq. 1, $\sum_{\mathcal M_i \sim p(\mathcal M)}$ is not standard, perhaps to rewrite it as $\sum_{\mathcal M_i} p(\mathcal M_i) \mathcal L_{\mathcal M_i}(\cdots)$


**Time Spent Reviewing:**

7 hours

---

> ### Author Response · Authors · 2021-08-10
> **Response to Reviewer uSF5**
>
> We thank the reviewer for useful feedback and helpful comments. FAMP indeed does not use warm-up period for high-level policy. This period and asynchronous updates are used in MLSH because it does not backpropagate through updates. We agree that this comparison can be made more clear in Section 4.
>
> Agents are initialized with 50% termination probabilities and randomly initialized sub-policies. While fixing sub-policies may lead to slower meta-learning, the adaptation should be much faster because the agent needs to adapt less parameters. Since our main goal is fast adaptation at meta-test time, we do not mind spending a bit more time with meta-training. Nevertheless, we see how the inclusion of a small ablation study, in which the effect of fixed sub-policies in the inner update is studied, can shed more light on the importance of decision factors. Since fixing of sub-policies does not require any significant changes in the algorithm design or code and this ablation was unanimously proposed by all reviewers, we will include this ablation study in the final version of the paper.
>
> Similarly, we agree with the reviewer that it would be useful to study the effect of fixed terminations on learning. However, adapting the algorithm to fixed terminations is not as straightforward and would require some more theoretical work to adjust the updates. We are thus not certain whether we would be able to include it in the paper.

---

### Decision · Program_Chairs · 2021-09-27

**Decision:**

Reject

**Comment:**

This paper proposes a new hierarchical RL algorithm for fast adaptation. Following the options framework, the proposed method learns a set of sub-policies, a termination function, and a high-level policy via MAML with DiCE. The results on a simple grid world (Taxi) and a continuous control navigation task (Ant Maze) show that the proposed method adapts to new tasks more efficiently than a hierarchical (MLSH) and non-hierarchical baselines (PPO/RL^2).

Although the idea is new and interesting, all of the reviewers agreed that the experiments are conducted only on simple domains. More crucially, while this paper emphasizes the importance of several ideas (e.g., learning termination function rather than using fixed-length options, fixing sub-policies during adaptation of high-level policy), there is no ablation study that justifies the claims. Thus, even though the results look encouraging, it is hard to understand the benefit of individual ideas. Therefore, this paper is not ready for publication at the moment, and I encourage the authors to conduct more comprehensive experiments for the future venue.